# Diagnostic intervention improved health-related quality of life among teenagers with food allergy

Linnéa Hedman[1]*, Åsa Strinnholm[2], Sven-Arne Jansson[1], Anna Winberg[2]

**1** Department of Public Health and Clinical Medicine, Section of Sustainable Health, The OLIN Unit, Umeå University, Umeå, Sweden, **2** Department of Clinical Sciences, Pediatrics, Umeå University, Umeå, Sweden

* linnea.hedman@norrbotten.se

**Data Availability Statement:** Data cannot be shared publicly because of legal and research ethical reasons. Data are available from Region Norrbotten (dataskyddsombud@norrbotten.se)

## Abstract

### Objectives

The aim was to examine if a diagnostic intervention set up to assess current food allergy to cow's milk, hen's egg, fish, or wheat among teenagers had an impact on generic and disease specific health-related quality of life (HRQL). The study compared HRQL scoring before and two years after the intervention, and in relation to age matched controls without reported food allergy.

### Methods

The study was performed within the Obstructive Lung Disease in Northern Sweden (OLIN) studies where a cohort study on asthma and allergic diseases among 8-year-old schoolchildren was initiated in 2006. At age 12 years, the 125/2612 (5%) children who reported allergy to cow's milk, hen's egg, fish, or wheat were invited to a diagnostic intervention including clinical examination, blood tests and evaluation by a pediatric allergist. Of 94 participants, 79 completed generic and disease specific HRQL questionnaires. Additionally, a random sample of 200 (62% of invited) children without food allergy from the OLIN cohort answered the generic HRQL questionnaire. The respondents of the HRQL questionnaires were re-examined two years later and 57 teenagers with and 154 without reported allergy participated.

### Results

There were no significant differences in generic HRQL scores between teenagers with and without reported food allergy at study entry, or after the intervention. Among those with reported food allergy, we found a significant improvement in disease specific HRQL after the intervention (mean values: 3.41 vs 2.80, p<0.001). Teenagers with only food allergy had better disease specific HRQL compared to those with one, two or three concomitant allergic diseases, both before and after the intervention. Children with only food allergy significantly improved their HRQL after the intervention, 1.84 vs. 2.87 (p<0.001) but this association was

upon reasonable request and after a confidentiality evaluation for researchers who meet the criteria for access to confidential data.

**Funding:** The study was funded by The Swedish Asthma-Allergy Foundation (AW) 2010013-K https://astmaoallergiforbundet.se/; The Swedish Heart-Lung Foundation (LH) 20140264 https://www.hjart-lungfonden.se/; VISARE Norr (LH) 370491 https://www.norrasjukvardsregionforbundet.se/halso-och-sjukvard/fou/visare-norr/; FoU - the State Government funding for Health Care research, AW: VLL-233971, ÅS: RV-158921;: RV-46700, LH: 484661, Insamlingsstiftelsen (AW) 223-27-10 https://www.aurora.umu.se/organisation-och-styrning/organisation/fakultetssidor/medicinsk-fakultet/utlysningar/insamlingsstiftelsen/; The Sven Jerring Foundation, AW 2015-12-08, ÅS: 20111208,20131209, 141208,151208,161215 https://jerringfonden.se/; The Kempe-Carlgrenska Foundation (ÅS) 20151111,20141114, 20130520 https://kc-fonden.se/; The Oskar foundation AW 20120116, 2011-01-18; ÅS 20140219, 20130121 http://www.oskarfonden.se/; The Anerska Foundation (AW) 223-2617-11 https://stiftelsemedel.se/stiftelsen-anerska-fonden/; The Kjerstin Hejdenberg Foundation (AW) 2013-12-05 https://astmaoallergiforbundet.se/forskning/sok-bidrag/kerstin-hejdenbergs-stipendium/; Ebba Danelius' Foundation (ÅS) 2014-11 https://swenurse.se/fonder-och-stipendier/projektmedel/ebba-danelius-stiftelse; Stiftelsen Samariten AW 2011-11-17; ÅS 20131122, 20141120, 2015-0014 https://www.stiftelsensamariten.se/; The Swedish Society for Asthma and Allergy Nursing (ÅS) 20150320 https://swenurse.se/sektionerochnatverk/astmaallergiochkolsjukskoterskeforeningen.4.62e5211e171ac42ce6c14897.html; The Mayflower Charity Foundation for Children (ÅS) 44 2014-01-27 https://majblomman.se/. The funders had no role in study design, data collection and analysis, decision to publish, or preparation of the manuscript.

**Competing interests:** The authors have declared that no competing interests exist.

not seen in children with one other allergic disorder (3.16 vs. 3.65, p = 0.121) or those with two or more allergic disorders (3.72 vs. 3.90, p = 0.148).

## Conclusion

The diagnostic intervention showed a long-term improvement of disease specific HRQL but not generic HRQL.

## Introduction

Food allergy (FA) is a disease with increasing prevalence and global health significance that affects social, emotional as well as financial well-being [1]. There is an estimated prevalence of immunoglobulin (Ig) E-mediated FA among children in Europe of 5–8% [1–3] but a recent European meta-analysis showed a pooled lifetime prevalence of reported FA of 19.9% [3–6]. When objective diagnostic methods are used in epidemiologic studies, the prevalence of self-reported FA usually exceeds that of verified allergy [3, 7, 8].

Studies on Health-Related Quality of Life (HRQL) in children and adolescents with FA show divergent results [9]. The different outcomes can often be related to differences in study design regarding age of study participants, type, and severity of symptoms, which HRQL instruments are used and if the study participants or their parents answered the questionnaires [9–12]. In many studies, a negative impact on HRQL is seen among children with severe symptoms of FA [11, 13, 14]. Diagnostic intervention with an oral food challenge has been associated with improved FA specific HRQL for the individual and a reduced burden on parents, independent of the challenge outcome [15]. Less is known about the impact on HRQL in children and adolescents with self-reported FA of other, less resource intensive diagnostic interventions [16].

We have previously compared HRQL in children from a population-based cohort of children aged 12 years with and without self-reported allergy to cow's milk, hen's egg, fish, or wheat and found no significant differences in generic HRQL between groups [13]. Children with reported symptoms to staple foods were invited to an assessment of current FA including structured interviews, clinical examination, blood tests and evaluation by a pediatric allergist. Study participants assessed as having a current FA were offered further investigation with a double-blind placebo-controlled food challenge (DBPCFC) series. If a current FA could be excluded, food reintroduction was recommended [8].

The objective of this study was to examine if a diagnostic intervention including structured interviews, clinical examination, blood tests and information given by a pediatric allergist had a long-term impact on generic and disease specific HRQL among teenagers with self-reported allergy to cow's milk, hen's egg, fish or wheat. The study compared HRQL scoring before the intervention and two years after, and in relation to age matched controls without FA.

## Materials and methods

### Study design and study sample

The study was performed within the Obstructive Lung Disease in Northern Sweden (OLIN) studies. In 2006, the prospective, population-based OLIN pediatric study II was initiated [17]. The parents of all children in first and second grade (age 7–8 years) in three municipalities in the county of Norrbotten, Sweden were invited to complete a questionnaire [18] about asthma

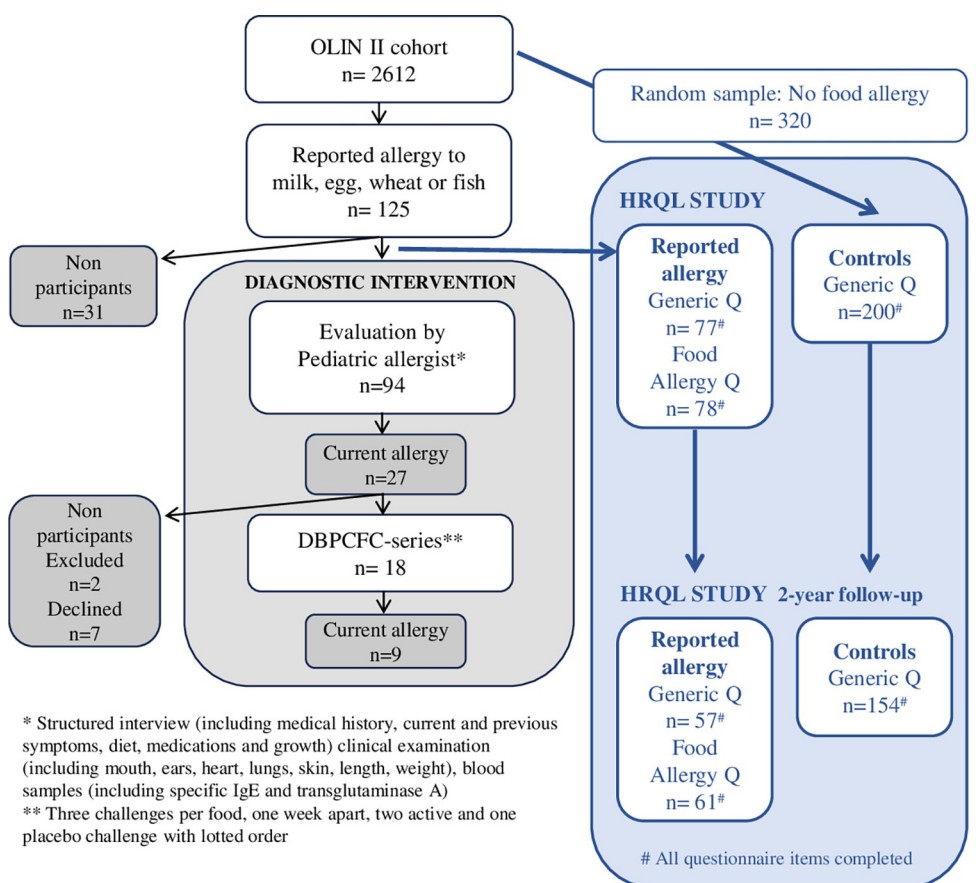

**Fig 1. Flow chart of the OLIN pediatric study II and recruitment of participants to the study on food allergy and health-related quality of life.** * Structured interview (including medical history, current and previous symptoms, diet, medications and growth) clinical examination (including mouth, ears, heart, lungs, skin, length, weight), blood samples (including specific IgE and transglutaminase A). ** Three challenges per food, one week apart, two active and one placebo challenge with lotted order.

and allergic diseases, including FA, and their related risk factors [5, 17] (Fig 1). The cohort study and the interventions have been approved by the Swedish Ethical Review Authority (Dnr 2005-157M; Dnr 09–206; Dnr 2010-247-31; Dnr 2011-34-32M). The parents provided written informed consent for their child to participate.

In February-May of 2010, when the children were 11–12 years of age, they were invited to a follow-up study and 2612 participated [5]. Based on self-reported questionnaire data, five percent (n = 125) of the children reported complete avoidance of cow's milk, hen's egg, fish and/or wheat due to self-perceived allergy. These children were invited to a clinical evaluation by a pediatric allergist (AW) and a specialized allergy nurse (ÅS). The diagnostic evaluation included a structured interview, clinical examination, and analyses of food specific IgE and transglutaminase A [8]. Of the participants, 27 (28%) were assessed as having a current FA but due to contraindications, two children were excluded from further evaluation. Of the remaining 25, 18 children (72%) participated in 20 DBPCFC-series. When a current FA could be excluded, food reintroduction was recommended [8].

During the diagnostic evaluation, children were asked to complete one generic and one disease specific HRQL. In total, 77 (84%) children with self-reported FA participated and completed all items in the generic questionnaire, while 78 completed all items in the disease

specific questionnaire. In addition, a random sample of 320 children without FA from the OLIN study II cohort was invited to answer the generic HRQL questionnaire, which was sent to them by mail. Of these, 200 children participated and completed all items in the question-naire [13].

In 2012, when the study participants were 14 years of age, a follow-up study was performed using the same generic and disease specific HRQL questionnaires. In total, 57 teenagers with and 154 teenagers without FA completed all items in the generic questionnaire. Furthermore, 61 teenagers answered all items in the disease specific questionnaire. Study participants who provided complete HRLQ data both at study entry and at the follow-up study were included in the analyses.

## HRQL questionnaires

The same generic and disease specific questionnaires were used both at study entry and at fol-low-up. The generic self-report Health-related Quality of Life Screening Instrument, KIDSC-REEN-52 measures 10 domains of health, Physical Well-being, Psychological Well-being, Moods and Emotions, Self-Perception, Autonomy, Parent Relation and Home Life, Financial Resources, Social Support and Peers, School Environment, and Social Acceptance and Bullying [19, 20]. All answers are given on a 5-point scale where 1 is worst HRQL and 5 is best HRQL. The scores were transformed into a scale from 0–100 for each domain where the population norm is a mean of 50 and standard deviation of 10. A higher score indicates a better HRQL [20].

The disease specific Food Allergy Quality of Life Questionnaire Teenager Form (FAQLQ-TF) consists of 28 questions on HRQL, each of which has corresponding closed-ended answers on a 7-point scale where 1 is best HRQL and 7 is worst HRQL [21]. By taking the mean value of the 28 questions, a total HRQL score is established. These questions were also divided into three domains: Allergen Avoidance and Dietary Restrictions (AADR), Risk of Accidental Exposure (RAE) and Emotional Impact (EI). A lower score indicates a better HRQL.

## Statistical analyses and data management

Statistical analyses were performed with the IBM Statistical Package for the Social Sciences (SPSS) Statistics 29. For research purposes, the data was accessed mainly from March until December 2022. The non-parametric Mann-Whitney U test or the Kruskal-Wallis test were used to assess differences in median between cases and controls before and after the interven-tion, respectively [22]. Comparison of mean HRQL before and after the intervention was ana-lyzed by Paired Samples T-test. A two-sided p-value <0.05 was considered as statistically significant. In order to link the participants' HRQL scores before and after the intervention, the authors had access to information that could identify individual participants during data collection.

## Results

### Basic characteristics of the study sample

At study entry, our study sample included 79 teenagers with self-reported allergy to cow's milk, hen's egg, fish, or wheat (59% girls), and 209 teenagers without FA (48% girls). Concomi-tant allergic conditions were more common among teenagers with than without self-reported FA: asthma 24.1% vs 6.2%, rhinitis 26.6% vs 6.2% and eczema 30.4 vs 10.5% (p< 0.001 for all) (Table 1).

**Table 1. Comparison of sex, family history of food allergy and allergic conditions between children with and without food allergy.**

|  | Food allergy n = 79 | Not food allergy n = 209 | p-value |
|---|---|---|---|
|  | % (n) | % (n) |  |
| Female sex | 55.7 (44) | 48.3 (101) | 0.124 |
| Family history of food allergy | 48.1 (38) | 21.1 (44) | <0.001 |
| Asthma | 24.1 (19) | 6.2 (13) | <0.001 |
| Rhinitis | 26.6 (21) | 6.2 (13) | <0.001 |
| Eczema | 30.4 (24) | 10.5 (22) | <0.001 |

## Generic HRQL before and after intervention

When estimating HRQL with the generic KIDSCREEN-52 questionnaire, we found no significant differences in median scores of HRQL between children with and without self-reported FA at study entry (Fig 2), or after the intervention (Fig 3). Analyses performed separately for boys and girls showed similar results and they are presented in S1 and S2 Tables. When comparing the change in mean HRQL scores for children with and without self-reported FA before and after intervention, all domains showed worsening of HRQL except for Social acceptance and bullying that had improved among teenagers without FA (Table 2). Overall, we found significantly decreased HRQL in two domains among children with self-reported FA and in seven domains among children without FA.

## Disease-specific HRQL before and after intervention

Overall disease specific HRQL and domain specific HRQL are presented in Fig 4. The overall HRQL mean score was 3.41 (SD 1.25) before the intervention, 3.42 (SD 1.38) for girls and 3.40

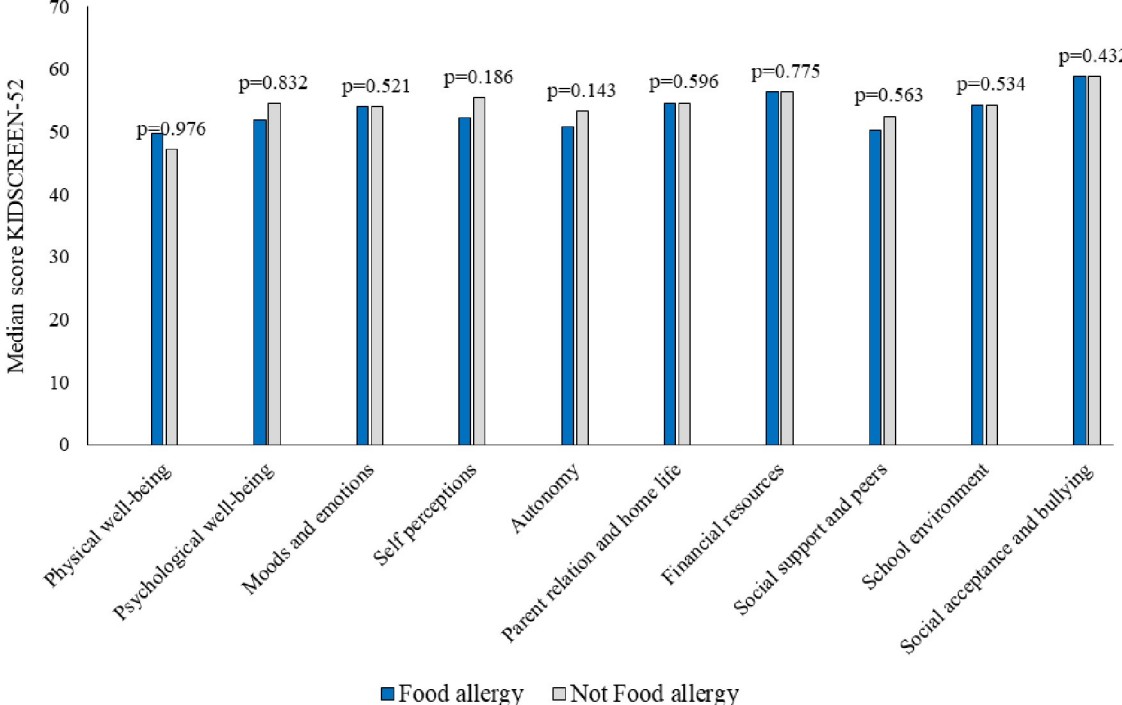

**Fig 2. Median scores in KIDSCREEN-52 domains among children with and without food allergy (FA), respectively, before intervention.** A higher score indicates better HRQL.

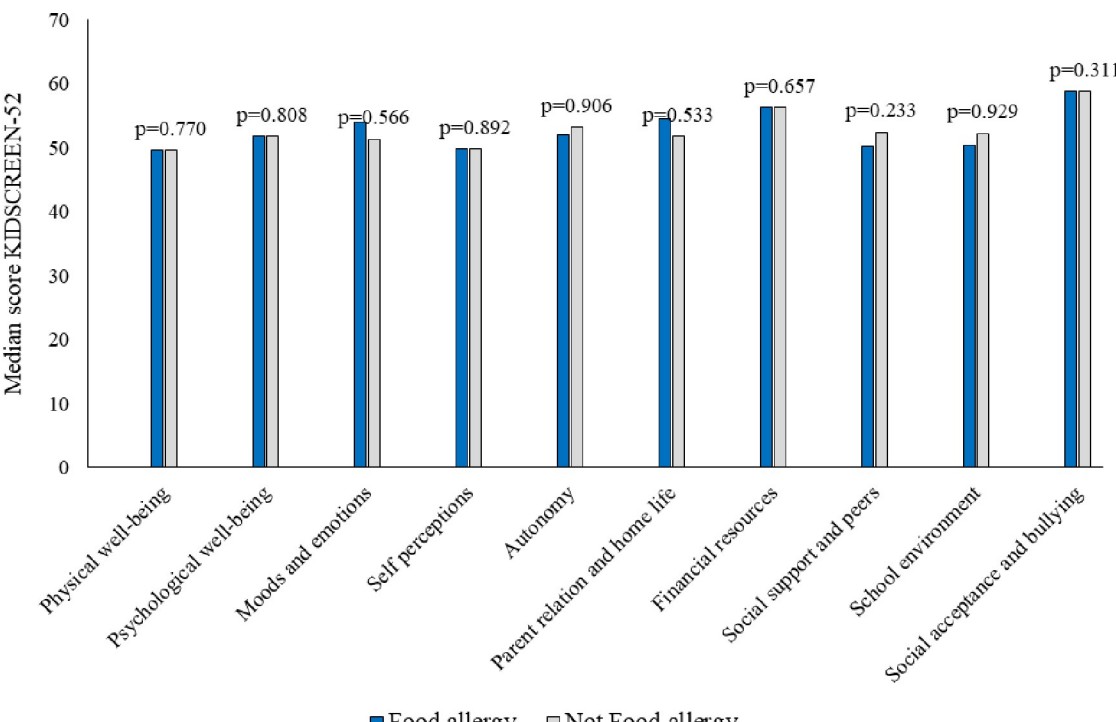

**Fig 3. Median scores in KIDSCREEN-52 domains among children with and without food allergy (FA), respectively, after intervention.** A higher score indicates better HRQL.

(SD 1.03) for boys. After the intervention, we found a significant improvement in overall HRQL for all domains. The mean overall HRQL improved significantly from 3.41 (SD 1.25) to 2.80 SD 1.41) after the intervention, p<0.001. For girls, the corresponding scores improved from 3.42 (SD 1.38) to 2.76 (SD 1.45) (p = 0.001) and for boys from 3.40 (SD 1.03) to 2.86 (SD 1.36) (p = 0.049).

In 19 out of 61 teenagers (31%) disease specific HRQL had decreased after the intervention. Among the teenagers with worsened HRQL, 26% had asthma compared with 17% among

**Table 2. Scores\* in KIDSCREEN-52 domains before and after intervention, among children with and without food allergy, respectively.**

| | Food allergy | | | Not food allergy | | |
|---|---|---|---|---|---|---|
| | Mean (SD) before | Mean (SD) after | p-value | Mean (SD) before | Mean (SD) after | p-value |
| Physical well-being | 46.9 (7.3) | 45.9 (7.8) | 0.292 | 46.6 (6.7) | 45.4 (6.5) | 0.040 |
| Psychological well-being | 53.4 (9.4) | 50.8 (11.0) | 0.102 | 53.3 (8.9) | 51.0 (9.1) | 0.002 |
| Moods and emotions | 54.5 (10.4) | 52.0 (10.8) | 0.091 | 55.6 (10.3) | 53.3 (10.6) | 0.015 |
| Self perceptions | 54.4 (9.6) | 52.0 (11.2) | 0.126 | 55.7 (9.6) | 51.2 (9.1) | <0.001 |
| Autonomy | 52.0 (10.4) | 54.2 (10.1) | 0.200 | 54.0 (8.4) | 53.7 (7.7) | 0.687 |
| Parent relation and home life | 54.6 (8.3) | 54.0 (8.9) | 0.617 | 55.3 (8.8) | 53.0 (8.8) | 0.004 |
| Financial resources | 56.1 (7.8) | 53.8 (9.0) | 0.043 | 55.7 (7.8) | 54.5 (9.1) | 0.110 |
| Social support and peers | 52.3 (9.6) | 50.1 (11.6) | 0.194 | 53.0 (9.3) | 52.5 (10.8) | 0.613 |
| School environment | 55.5 (9.6) | 52.2 (8.9) | 0.017 | 55.1 (10.0) | 52.6 (8.8) | 0.003 |
| Social acceptance and bullying | 53.1 (8.7) | 54.9 (7.4) | 0.137 | 53.8 (8.9) | 56.0 (6.4) | 0.003 |

\*A higher score indicates better HRQL. SD: standard deviation

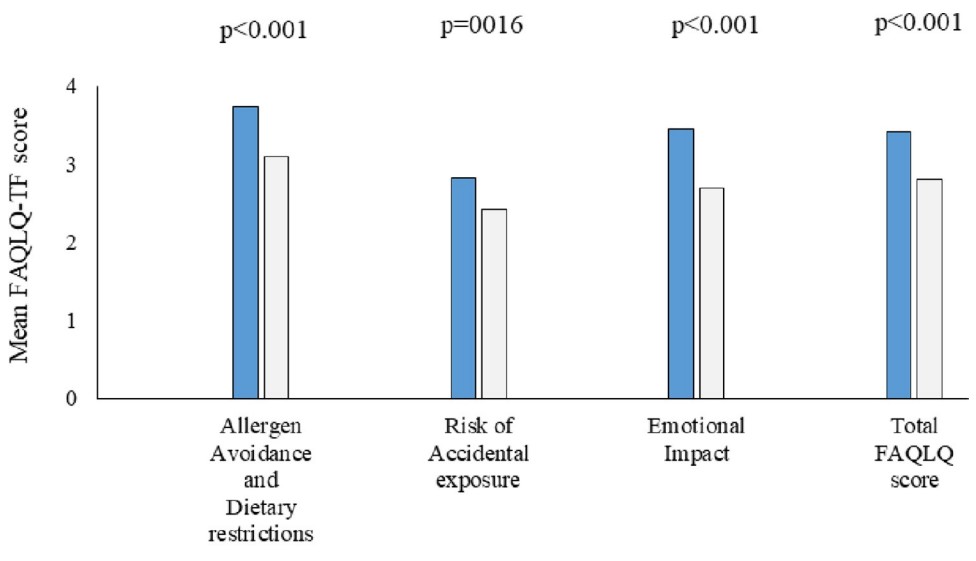

**Fig 4. Mean scores in FAQLQ-TF domains before and after intervention.** A lower score indicates better HRQL (n = 61).

those with improved HRQL (p = 0.489). The same trend was found for rhinitis, 32% vs. 24% (p = 0.543).

Teenagers with FA but no asthma, rhinitis or eczema had better disease specific HRQL compared to those with one, two or three other allergic diseases, both before and after the intervention (Fig 5). When comparing the mean disease specific HRQL scores before and after

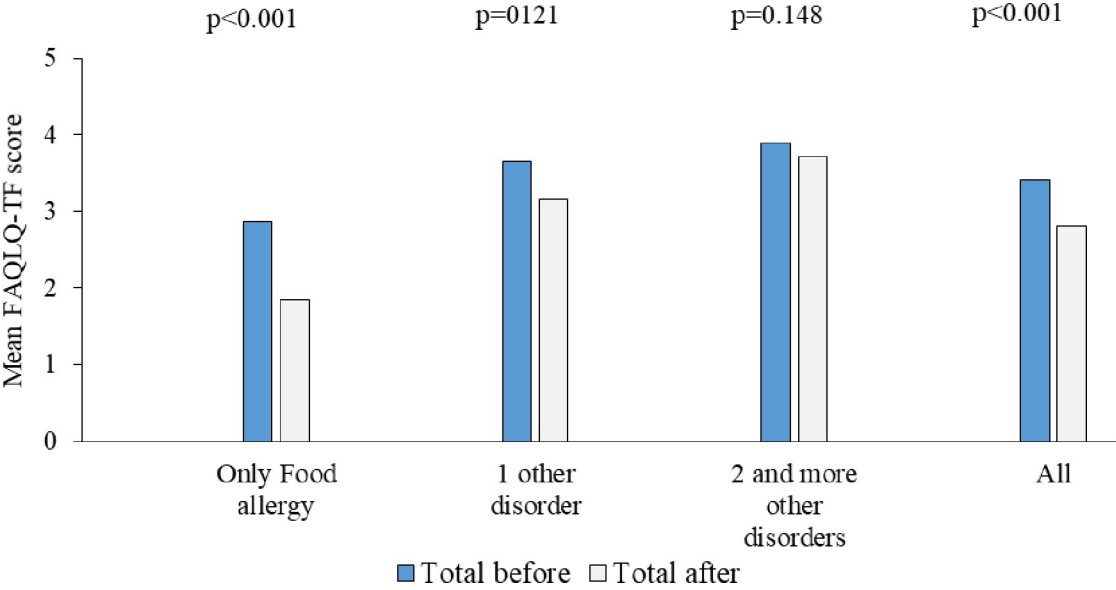

**Fig 5. Mean scores in FAQLQ-TF for children according to number of allergic diseases (asthma, rhinitis, and/or eczema).** A lower score indicates better HRQL.

the intervention by number of allergic diseases, we found that children with only FA significantly improved their HRQL after the intervention, 2.87 (SD 1.02) vs. 1.84 (SD 0.77) (p<0.001). No significant differences in mean disease specific HRQL scores were found in children with one, two or three other diseases, respectively.

In total, 18 teenagers participated in DBPCFC-series. These teenagers had higher scores (worse HRQL) in the disease specific questionnaire before the intervention compared to those who only participated in the clinical examination, 3.98 (SD 1.55) and 3.18 (SD 1.03), respectively (p = 0.020). Both groups improved the total scores significantly after the intervention but the difference between groups remained after the intervention, 3.34 (SD 1.56) compared to 2.57 (SD 1.29) (p = 0.053).

## Lost to follow-up

A non-response analysis was performed to investigate if those who did not participate in the follow-up after the intervention differed from those who participated both at study entry and at follow-up. We found no significant differences in basic characteristics such as sex, family history of FA and concomitant allergic diseases between groups. In addition, no significant differences were found in the scores of the generic questionnaire at baseline between those who participated in the two-year follow-up and those who did not.

## Discussion

This study compared HRQL among teenagers with and without self-reported allergy to cow's milk, hen's egg, fish, or wheat in a large population-based cohort two years after they participated in a diagnostic intervention. The intervention included structured interviews, clinical examination, blood tests and evaluation by a pediatric allergist. Our results show that the diagnostic intervention had a long-term impact on disease specific HRQL. In contrast, no impact on generic HRQL was shown after the intervention.

HRQL is often defined as self-perceived health status [23]. Experiences of HRQL differ between people and can be influenced by socioeconomic status [24], gender, age [25, 26], and self-perceived disease [13, 27, 28]. A strength of our study is that both generic and disease specific HRQL were investigated. According to European norm [20], our study participants with self-reported FA had a good generic HRQL both before and after the diagnostic intervention and compared to teenagers without FA [13]. We found that age had an impact on HRQL in our study. The mean generic HRQL scores for teenagers with and without self-reported FA had worsened at our two-year follow-up for all domains except two. This is in line with a large European study that showed better HRQL in children than in adolescents [26]. In that study, they also found a similar HRQL among girls and boys at a younger age, but a larger decrease in girls than boys as they got older [26]. No significant gender differences could, however, be found in the analyses of generic HRQL in our study.

Disease specific HRQL questionnaires focus on a particular disease e.g., FA and are designed to have a good sensitivity for HRQL in subjects with the target disease. In our study, disease specific HRQL showed improvement in overall HRQL for all domains two years after the diagnostic intervention and was significant for both girls and boys. In calculations based on all study participants as well as on girls and boys respectively, the difference in disease specific HRQL scores before and after the intervention exceeded 0.5, which is considered a clinically significant difference [29]. Diagnostic intervention by oral food challenges can improve disease specific HRQL independent of the challenge outcome [15] and often with greater improvements after a negative challenge [30]. The improvement in HRQL following an oral food challenge is thought to be caused by a confirmed or ruled out FA diagnosis and better

knowledge about the disease and its treatment [30]. This has been highlighted in previous interview studies performed by our group [14, 31].

However, less is known about the effect of other diagnostic interventions on HRQL in children and adolescents with self-reported FA [32]. Since oral food challenges in health care settings are time and resource consuming [33] and perceived FA is prevalent among children and adolescents [4, 5, 33], there is a need for more easily available methods for first line evaluations. In our study, most of our study participants only took part in the pediatric allergist evaluation and were thereafter recommended reintroduction of the previously eliminated food. Only those assessed as having a current FA were invited to further examination with a DBPCFC-series [8]. Our findings suggest that in a population-based setting, a less resource intensive intervention such as a clinical evaluation could be a sufficient tool for avoiding unnecessary elimination diets and negative impact on HRQL in children with self-perceived allergy to foods.

Our study participants who were invited to further evaluation with DBPCFC reported worse disease specific HRQL as well as more severe and more recent symptoms of FA compared to those who only participated in the clinical evaluation [8, 13]. Despite clinically significant improvement in HRQL [29] in both intervention groups, the differences between groups remained at the two-year follow-up. This is in line with other studies where a larger negative impact on HRQL often is seen among children with severe symptoms of FA [11, 12, 34]. Objective disease severity is, however, not the only factor that can affect HRQL [32, 35]. In a European study, self-perceived disease severity was a predictor of impaired HRQL in both adults and children. Other predictors were gender, type of symptoms and allergy to fish and milk in adults and country of origin and allergy to peanut and soy in children [35].

While most teenagers with self-reported allergy to cow's milk, hen's egg, fish, or wheat in our study improved their disease specific HRQL, almost a third of our study participants had a worse HRQL at the two-year follow-up. There was no difference in the distribution between girls and boys between this group and those who improved their HRQL after the intervention. Teenagers with lower scores did, however, more often report allergic conditions like asthma, rhinitis and eczema compared to those with higher HRQL scores. Allergic diseases often overlap [1, 17] and although studies about the association between allergic multi-morbidity and changes in HRQL during adolescence are scarce [16], an association between clinical or general well-being among adolescents and number of allergic conditions has been reported [11, 16]. Furthermore, both before and after our study intervention, teenagers with self-reported FA but without asthma, rhinitis or eczema, had better disease specific HRQL compared to those with one, two or three additional allergic diseases. When comparing the mean disease specific HRQL scores before and after the intervention by number of allergic diseases, we also found that children with only FA significantly improved their HRQL after the intervention, while no significant differences in mean disease specific HRQL scores were found in children with one, two or three additional allergic conditions, respectively.

Major strengths of our study are the longitudinal population-based design and the use of generic as well as disease specific HRQL instruments. Our study population was well defined, which allowed analyses of factors other than self-reported FA that could affect HRQL, and the population-based cohort study allowed recruitment of age-matched controls without FA from the same catchment area. A study weakness is that we had some lost to follow-up in the generic HRQL questionnaire, especially among the controls. However, a non-response analysis showed no differences in basic characteristics such as sex, family history of FA and concomitant allergic diseases nor in the scores of the first generic HRQL questionnaire between teenagers who participated in the two-year follow-up and those who did not. Another limitation is the sample size of the group with FA, which lacked statistical power for some analyses. For

instance regarding mean values of the generic HRQL questionnaire before and after the intervention where corresponding mean difference yielded statistically significant results for the group without FA.

In this large population-based study, we found an improved disease specific HRQL in teenagers with self-reported allergy to cow's milk, hen's egg, fish, or wheat two years after a diagnostic intervention including structured interviews, clinical examination, blood tests and evaluation by a pediatric allergist. The improvement in disease specific HRQL was seen despite of a worsened generic HRQL in teenagers with as well as without reported FA. Our results pinpoint the importance of follow-up and evaluation of children with self-perceived allergy to foods and suggests that in many cases, a clinical evaluation might be sufficient to avoid unnecessary elimination diets and a negative impact on HRQL. Our results also show the importance of disease specific HRQL instruments for the assessment of HRQL in children with FA over time and in relation to interventions.

## Supporting information

**S1 Table. Scores in KIDSCREEN-52 domains before intervention among girls and boys, with and without food allergy, respectively.**
(DOCX)

**S2 Table. Scores in KIDSCREEN-52 domains after intervention among girls and boys, with and without food allergy, respectively.**
(DOCX)

**S1 File.**
(PDF)

**S2 File.**
(DOCX)

## Acknowledgments

Professor Eva Rönmark is acknowledged for study conception and scientific support.

## Author Contributions

**Conceptualization:** Linnéa Hedman, Åsa Strinnholm, Anna Winberg.

**Data curation:** Åsa Strinnholm.

**Formal analysis:** Sven-Arne Jansson.

**Funding acquisition:** Linnéa Hedman, Åsa Strinnholm, Anna Winberg.

**Investigation:** Åsa Strinnholm, Anna Winberg.

**Methodology:** Åsa Strinnholm, Anna Winberg.

**Project administration:** Åsa Strinnholm.

**Resources:** Linnéa Hedman.

**Supervision:** Linnéa Hedman.

**Validation:** Anna Winberg.

**Visualization:** Sven-Arne Jansson.

**Writing – original draft:** Linnéa Hedman, Sven-Arne Jansson, Anna Winberg.

**Writing – review & editing:** Linnéa Hedman, Åsa Strinnholm, Sven-Arne Jansson, Anna Winberg.

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
