## [Decision Letter · Decision Letter 0]

16 Oct 2023

PONE-D-23-19870Diagnostic intervention improved health-related quality of life among teenagers with food allergyPLOS ONE

Dear Dr. Hedman,

Thank you for submitting your manuscript to PLOS ONE. After careful consideration, we feel that it has merit but does not fully meet PLOS ONE’s publication criteria as it currently stands. Therefore, we invite you to submit a revised version of the manuscript that addresses the points raised during the review process.

We look forward to receiving your revised manuscript.

Kind regards,

Dong Keon Yon, MD, FACAAI, FAAAAI

Academic Editor

PLOS ONE

Journal Requirements:

Additional Editor Comments:

Please address the excellent comments from the reviewers.

#1. SPSS 24 -> this is too old version of statistical program. Please use v26 or v27.

#2. The non-parametric Mann-Whitney U test or the Kruskal-Wallis test were used to assess differences in median between cases and controls before and after the intervention, respectively -> Please cite the statistical guideline (DOI: https://doi.org/10.54724/lc.2022.e1).

#3. A p-value <0.05 was considered as -> A two-sided p-value <0.05 was considered as

This is an excellent paper.

Reviewers' comments:

Reviewer's Responses to Questions

**Comments to the Author**

1. Is the manuscript technically sound, and do the data support the conclusions?

Reviewer #1: Yes

Reviewer #2: Partly

Reviewer #3: Partly

2. Has the statistical analysis been performed appropriately and rigorously? 

Reviewer #1: Yes

Reviewer #2: Yes

Reviewer #3: Yes

3. Have the authors made all data underlying the findings in their manuscript fully available?

Reviewer #1: No

Reviewer #2: No

Reviewer #3: No

4. Is the manuscript presented in an intelligible fashion and written in standard English?

Reviewer #1: Yes

Reviewer #2: Yes

Reviewer #3: Yes

5. Review Comments to the Author

Reviewer #1: In this population-based study, generic and disease specific HRQL was compared among teenagers with and without self-reported food allergy before and after a diagnostic intervention.

The study is well-designed, clearly written and has several strengths as are well-described by the authors with important clinical implications. I only have a few minor suggestions for improvement:

Abstract line 43: the number 1.84 vs. 2.87 are difficult to interpret without reading the whole paper (what is compared to what?). Are they changes in HRQL among those with only food allergy vs additional allergies or are they before and after values among those with only food allergies and in that case, what were the corresponding values for those with additional allergies?

There are a lot of numbers in the methods section to keep track of and sometimes difficult to follow. This could perhaps be written even more clearly for example include the participation rate at the clinical evaluation among those with self-perceived allergy. The flow-chart is useful, but it looks like the first HRQL assessment are performed after the DBPCFC.

Line 181 and 182: should decreased be “worsened”? (increased values?)

Line 163-164: “Overall, we found significantly decreased HRQL in two domains among children with self-reported FA in seven domains among children without FA” This is probably due to less power in the FA group, which may be added as a limitation in the discussion section.

Reviewer #2: Hedman L, et al. conducted a clinical study to evaluate the impact of diagnostic intervention for health-related quality of life (HRQL) in teenagers with food allergy. It demonstrated that the diagnostic intervention showed a long-term improvement of disease specific HRQL but not generic HRQL. However, I have several concerns.

1. I did not understand exactly what a Diagnostic Intervention is. A detailed description of the structured interview and each test and its frequency should be included in the Materials and Methods section.

2. In summarizing the data, it is easier for the readers to understand if not only the mean or median value is shown, but also a measure of variability. The mean should be accompanied by the standard deviation and the median by the range or interquartile range.

3. Also, for data with correspondence, be sure to show summary statistics of before and after changes.

Just minor concern

1. In the abstract, HRQL should be spelled out.

Reviewer #3: Dear author congratulation on your work.

The article addresses an important clinical dilemma. the ideas and methods were good. I think the study sample was too small to achieve the goals and outcome of the articles based on your analysis plan. Moreover, some information about the details of physical examination and test were done is not clearly stated on the methodology section.

I guess if you handle this issues in the future this article will add lots for the clinical practice in the field of food allergy.

Thanks

6. PLOS authors have the option to publish the peer review history of their article (what does this mean?). If published, this will include your full peer review and any attached files.

Reviewer #1: No

Reviewer #2: No

Reviewer #3: No

---

## [Author Response · Author response to Decision Letter 0]

24 Nov 2023

PONE-D-23-19870

Diagnostic intervention improved health-related quality of life among teenagers with food allergy

PLOS ONE

Dear Editors of PLOS ONE,

We are grateful for the constructive comments from the reviewers and for your decision to invite us to submit a revised manuscript. The comments and questions from the reviewers have been answered point by point, and the manuscript has been revised accordingly. The changes in the manuscript are highlighted by using trach changes. We believe that the manuscript has improved and we hope it merits for publication in PLOS ONE.

On behalf of all co-authors,

Linnea Hedman, Head of The OLIN studies

Associate professor, Department of Public Health and Clinical Medicine

Umeå University, Sweden

Authors’ response: We have reviewed the style requirements and made some revisions regarding naming of files.

Authors’ response: We have ensured that the information in Funding information is correct. I am afraid that we misunderstood the instructions to authors and did not provide a financial disclosure in the manuscript. Now we have added the list of funders also in the manuscript and made sure that they match the list in the submission form.

Additional Editor Comments:

Please address the excellent comments from the reviewers.

#1. SPSS 24 -> this is too old version of statistical program. Please use v26 or v27.

Authors’ response: We have re-made the analyses in version 29 of IBM SPSS statistics. There were no differences in the results after using the newer version of the software. 

#2. The non-parametric Mann-Whitney U test or the Kruskal-Wallis test were used to assess differences in median between cases and controls before and after the intervention, respectively -> Please cite the statistical guideline (DOI: https://doi.org/10.54724/lc.2022.e1).

Authors’ response: We thank the Editor for the suggestion and we have added the reference to the manuscript.

#3. A p-value <0.05 was considered as -> A two-sided p-value <0.05 was considered as

This is an excellent paper.

Authors’ response: We thank the reviewer for this suggestion and we have revised the manuscript accordingly. We also thank for the positive assessment of our paper.

Reviewers' comments:

Reviewer's Responses to Questions

Comments to the Author

1. Is the manuscript technically sound, and do the data support the conclusions?

Reviewer #1: Yes

Reviewer #2: Partly

Reviewer #3: Partly

2. Has the statistical analysis been performed appropriately and rigorously?

Reviewer #1: Yes

Reviewer #2: Yes

Reviewer #3: Yes

3. Have the authors made all data underlying the findings in their manuscript fully available?

Reviewer #1: No

Reviewer #2: No

Reviewer #3: No

4. Is the manuscript presented in an intelligible fashion and written in standard English?

Reviewer #1: Yes

Reviewer #2: Yes

Reviewer #3: Yes

5. Review Comments to the Author

Reviewer #1: In this population-based study, generic and disease specific HRQL was compared among teenagers with and without self-reported food allergy before and after a diagnostic intervention.

The study is well-designed, clearly written and has several strengths as are well-described by the authors with important clinical implications. I only have a few minor suggestions for improvement:

Authors’ response: We thank the reviewer for the overall positive assessment of our paper.

Abstract line 43: the number 1.84 vs. 2.87 are difficult to interpret without reading the whole paper (what is compared to what?). Are they changes in HRQL among those with only food allergy vs additional allergies or are they before and after values among those with only food allergies and in that case, what were the corresponding values for those with additional allergies?

Authors’ response: We thank the reviewer for noticing this unclear sentence. The mean values are the before and after values for children with only food allergy. We have revised the sentence and added the mean values for those with more allergic disorders.

There are a lot of numbers in the methods section to keep track of and sometimes difficult to follow. This could perhaps be written even more clearly for example include the participation rate at the clinical evaluation among those with self-perceived allergy. The flow-chart is useful, but it looks like the first HRQL assessment are performed after the DBPCFC.

Authors’ response: Regarding the flow chart, it has been revised for clarity. Regarding the presentation of the cohort, we have revised the text in the method section and omitted some of the presented numbers as suggested.

Line 181 and 182: should decreased be “worsened”? (increased values?)

Authors’ response: We thank the reviewer for the suggestion and we have clarified that the word should indeed be ‘worsened’.

Line 163-164: “Overall, we found significantly decreased HRQL in two domains among children with self-reported FA in seven domains among children without FA” This is probably due to less power in the FA group, which may be added as a limitation in the discussion section.

Authors’ response: We thank the reviewer for the suggestion and have added this information in the strength and limitation section of the discussion, page 16. 

Reviewer #2: Hedman L, et al. conducted a clinical study to evaluate the impact of diagnostic intervention for health-related quality of life (HRQL) in teenagers with food allergy. It demonstrated that the diagnostic intervention showed a long-term improvement of disease specific HRQL but not generic HRQL. However, I have several concerns.

1. I did not understand exactly what a Diagnostic Intervention is. A detailed description of the structured interview and each test and its frequency should be included in the Materials and Methods section.

Authors’ response: In line with a suggestion from Reviewer #1, we have revised the flow chart (figure 1) and also added information about the diagnostic intervention in the figure.

2. In summarizing the data, it is easier for the readers to understand if not only the mean or median value is shown, but also a measure of variability. The mean should be accompanied by the standard deviation and the median by the range or interquartile range.

3. Also, for data with correspondence, be sure to show summary statistics of before and after changes.

Authors’ response to point 2 and 3: We thank the reviewer for the suggestion and we have added standard deviation to the mean values presented in the tables and in the text.

Just minor concern

1. In the abstract, HRQL should be spelled out.

Authors response: We thank the reviewer for noticing, we have revised the abstract accordingly.

Reviewer #3: Dear author congratulation on your work.

The article addresses an important clinical dilemma. the ideas and methods were good. I think the study sample was too small to achieve the goals and outcome of the articles based on your analysis plan. Moreover, some information about the details of physical examination and test were done is not clearly stated on the methodology section.

I guess if you handle this issues in the future this article will add lots for the clinical practice in the field of food allergy.

Thanks

Authors response: We thank the reviewer for the feedback. Also Reviewer 1 and 2 brought up the sample size and the description of the physical examination and we have made revisions in the discussion and in the method section accordingly.

6. PLOS authors have the option to publish the peer review history of their article (what does this mean?). If published, this will include your full peer review and any attached files.

Do you want your identity to be public for this peer review? For information about this choice, including consent withdrawal, please see our Privacy Policy.

Reviewer #1: No

Reviewer #2: No

Reviewer #3: No

---

## [Decision Letter · Decision Letter 1]

18 Dec 2023

Diagnostic intervention improved health-related quality of life among teenagers with food allergy

PONE-D-23-19870R1

Dear Dr. Hedman,

We’re pleased to inform you that your manuscript has been judged scientifically suitable for publication and will be formally accepted for publication once it meets all outstanding technical requirements.

Kind regards,

Dong Keon Yon, MD, FACAAI, FAAAAI

Academic Editor

PLOS ONE

Additional Editor Comments (optional):

This is an excellent paper.

Reviewers' comments:

Reviewer's Responses to Questions

**Comments to the Author**

1. If the authors have adequately addressed your comments raised in a previous round of review and you feel that this manuscript is now acceptable for publication, you may indicate that here to bypass the “Comments to the Author” section, enter your conflict of interest statement in the “Confidential to Editor” section, and submit your "Accept" recommendation.

Reviewer #1: All comments have been addressed

2. Is the manuscript technically sound, and do the data support the conclusions?

Reviewer #1: Yes

3. Has the statistical analysis been performed appropriately and rigorously? 

Reviewer #1: Yes

4. Have the authors made all data underlying the findings in their manuscript fully available?

Reviewer #1: No

5. Is the manuscript presented in an intelligible fashion and written in standard English?

Reviewer #1: Yes

6. Review Comments to the Author

Reviewer #1: Thank you for addressing all my comments and concerns. I have no further suggestions for improvement.

7. PLOS authors have the option to publish the peer review history of their article (what does this mean?). If published, this will include your full peer review and any attached files.

Reviewer #1: No

---

## [Editor Report · Acceptance letter]

2 Jan 2024

PONE-D-23-19870R1 

PLOS ONE

Dear Dr. Hedman, 

I'm pleased to inform you that your manuscript has been deemed suitable for publication in PLOS ONE. Congratulations! Your manuscript is now being handed over to our production team.

Kind regards, 

on behalf of

Dr. Dong Keon Yon 

Academic Editor

PLOS ONE